# Inter-Observer Repeatability of Indicators of Consciousness after Waterbath Stunning in Broiler Chickens

**DOI:** 10.3390/ani12141800

**Published:** 2022-07-13

**Authors:** Alexandra Contreras-Jodar, Aranzazu Varvaró-Porter, Virginie Michel, Antonio Velarde

**Affiliations:** 1Animal Welfare Program, Institute of Agrifood Research and Technology (IRTA), 17121 Monells, Spain; alexandra.contreras@irta.cat (A.C.-J.); aranzazu.varvaro@irta.cat (A.V.-P.); 2Direction of Strategy and Programmes, French Agency for Food, Environmental and Occupational Health & Safety (ANSES), 14 Rue Pierre et Marie Curie, 94701 Maisons-Alfort, France; virginie.michel@anses.fr

**Keywords:** inter-observer repeatability, prevalence, animal-based indicators, state of consciousness, slaughterhouse, waterbath stunning, broiler

## Abstract

**Simple Summary:**

Waterbath stunning is intended to induce unconsciousness until death occurs due to bleeding in poultry. However, it is not always effective. For this reason, in order to protect their welfare, it is mandatory in the European Union that the state of consciousness of broiler chickens is monitored at the exit of the waterbath, and that they do not regain consciousness before death. Ineffectively stunned birds can be re-stunned using back-up methods to avoid unnecessary pain, stress and suffering. One of the main challenges in monitoring the state of consciousness in broiler chickens after waterbath stunning is the selection of animal-based indicators ensuring consistency among assessments. The indicators should be valid, feasible and repeatable. However, only the validity and feasibility have been reported. Thus, the main goal of this research was to assess the repeatability of the most valid and feasible indicators of the state of consciousness after waterbath stunning in broilers both before bleeding (tonic seizure, breathing, spontaneous blinking and vocalization) and during bleeding (wing flapping, breathing, spontaneous swallowing and head shaking). This study proposes a refined list of indicators that could be used to assess the consciousness of broiler chickens in commercial slaughterhouses.

**Abstract:**

This study evaluated the prevalence and tested the inter-observer repeatability of the most valid and feasible animal-based indicators of the state of consciousness after waterbath stunning in broilers before bleeding (tonic seizure, breathing, spontaneous blinking and vocalization) and during bleeding (wing flapping, breathing, spontaneous swallowing and head shaking). In addition, correlations among them were computed to better understand their relationship and offer insights into the reliability of such indicators. This was aimed at proposing a refined list of indicators that could be used in commercial slaughterhouses to ensure consistent assessments. This study compared the assessments of three observers of 5241 broilers from 19 batches in six different slaughterhouses. Inter-observer repeatability was assessed through the combination of the crude percentage of agreement and the Fleiss’ kappa coefficient and interpretation. Before bleeding, the results led us to recommend assessing breathing over spontaneous blinking and vocalizations and to neglect tonic seizure in commercial conditions. During bleeding, the recommended indicators are breathing, wing flapping and head shaking while spontaneous swallowing can be neglected.

## 1. Introduction

Two main types of stunning systems are used commercially in broiler chickens: electrical waterbath stunning (WBS) and controlled atmosphere stunning (CAS). WBS is by far the most frequently applied method worldwide [1,2]. It consists in hanging birds by their legs in metal shackles on a moving line which takes them to a waterbath, where they are immersed up to the base of the wings in electrified water, where they are stunned. The contact of the head and neck with the water completes the electric circuit between the water (positive electrode) and the shackle line, which acts as the earth or negative bar electrode, so that the electric current passes through the bird’s head and body [3]. If sufficient current passes through the brain, its normal function is disrupted, and the animal is immediately rendered unconscious. This unconsciousness is a result of temporary or permanent damage to normal brain function, and the individual is unable to perceive and respond to external stimuli, including pain [4].

Although WBS is intended to induce unconsciousness until death occurs due to bleeding, it is not always effective [5]. For this reason, in order to protect the welfare of birds, it is mandatory in the European Union that the state of consciousness is monitored at the exit of the waterbath (WB) stunner and that the animals do not regain consciousness before death. Thus, ineffectively stunned birds can be re-stunned using back-up methods to avoid causing them unnecessary pain, stress and suffering.

One of the main challenges in monitoring the state of consciousness in broiler chickens after WBS is the selection of animal-based indicators (ABIs) ensuring the consistency of assessments. Recording electroencephalogram (EEG) is the most objective available method to ascertain the induction and maintenance of unconsciousness following stunning in broiler chickens [6]. In EEG analysis, the occurrence of a profoundly suppressed isoelectric EEG as well as epileptiform activity is normally associated with loss of consciousness [7,8,9]. However, at present, EEG can only be used under laboratory conditions. For this reason, under commercial conditions, the alternative is to combine the two main categories of indicators: resource-based indicators (RBIs) and animal-based indicators (ABIs). Traditionally, welfare assessments have focused on RBI assessments, such as on the key electrical parameters used and the minimum time that the birds spent submerged in the waterbath; these have been linked to the efficiency of stunning and set down in legislation. It was considered that if the resources were appropriate, animal welfare was guaranteed. However, RBIs may not reflect high welfare standards since, although the key electrical parameters established by the legislation are used, not all birds are successfully rendered unconscious and some may be ineffectively stunned or recover consciousness before death. Thus, animal welfare assessments based on an integrated approach in which both RBI and ABI are taken into consideration are necessary [10,11]. ABIs can more directly reflect the state of consciousness. Indeed, the results of the RBI indicate the risk of ineffective stunning but not the actual state of consciousness of each animal [12].

Hence, ABIs should meet three requirements, namely validity, feasibility and repeatability, in order to be relevant for assessing the state of consciousness. Validity tells us the extent to which an indicator is meaningful in terms of providing information about the presence of conscious birds, whereas feasibility refers to the applicability to different WBS equipment, slaughterhouse (SH) designs and different line speeds. Repeatability tells us the extent to which results are largely the same when the same observer repeats assessments, or the agreement between two or more observers after they have received reasonable training. If inter-observer repeatability is poor, then the indicator is probably inappropriate for welfare assessments.

The validity of the ABIs for the assessment of the state of consciousness in poultry was correlated with EEG outputs [7,13,14]. However, it was not until 2013 that EFSA [15] carried out the first step toward refining the perceived validity (i.e., sensitivity) and feasibility considered by experts in this field. In addition, they pointed out that birds should be monitored at two different stages of the slaughter line: before and during bleeding. Hence, a shorter list of ABIs was recommended according to the stage (with the highest validity and feasibility), with other parameters being seen as optional. However, inter-observer repeatability of any of these ABIs has not been assessed yet.

In the present study, the main goal is to evaluate the inter-observer repeatability of the four most valid and feasible ABIs according to EFSA [15], both before (i.e., tonic seizure, breathing, spontaneous blinking and vocalization) and during bleeding (i.e., wing flapping, breathing, spontaneous swallowing and head shaking) in different commercial slaughterhouses and with different batches of broilers. In addition, the prevalence of the outcomes of the ABIs are calculated to determine which are more prone to occur in cases of ineffective stunning and to identify correlations among them in terms of their reliability and relationship with the level of consciousness. From those observations, a reduced list of the most relevant ABIs is proposed for use in assessments of the consciousness of broiler chickens in commercial slaughterhouses, ensuring consistency between observers.

## 2. Materials and Methods

### 2.1. Selection of Slaughterhouses and Animals

Six commercial broiler chicken SH in France and Spain, equipped with WBS, were selected. Selections of the SH were carried together with the official veterinary services in order include a certain diversity in terms of the size of the plant, key electrical parameters, chicken genotypes and line speed. Each slaughterhouse was assigned a number (SH 1 to 6).

### 2.2. Description of the Slaughterhouses and Waterbath Stunning Systems

Substantial variation of age, design and construction of the SH was observed. The main characteristics are shown in Table 1.

In all SH, broilers were individually hung upside down by the legs on moving shackles of a slaughter line and stunned by immersion of the head in an electrified waterbath. None of them had adjustable shackles for different weights/sizes of the broiler legs. The height of the waterbath was adjusted according to the size of the birds to facilitate the immersion of all birds up to the base of their wings. Line speed was not measured in situ, but was reported by the food business operators and the official veterinary service. A digital control panel monitored the electrical parameters applied (i.e., total current passing through the waterbath, voltage and frequency) in all SH. The automatically recorded electrical parameters were obtained from the SH but were not measured and verified. The average values of current per animal were calculated by dividing the total current amount passing through the waterbath by the number of birds simultaneously in the water. The electrical waveform was sine alternating current in all SHs. The bleeding procedure differed among SHs: two did manual bleeding by cutting the carotids by an oropharynx incision (SH-1 and 6); two did it automatically (automatic neck cutter; SH-2 and 3); and two did a combination of automatic and manual (SH-4 and 5), as the automatic neck cutter only sectioned one of the carotids, and operators manually sectioned the second afterwards. Slaughter line speeds ranged from 200 to 10,500 birds/h. As the level of noise was not considered to interfere with the assessment of vocalizations in any of the SHs, the intensity of sound (dB) was not measured.

### 2.3. Assessment of the Consciousness

#### 2.3.1. Observers

The assessment of the effectiveness of stunning was carried out by three trained observers. Each observer was named with a letter (A to C). An additional person randomly selected and identified the birds to be assessed by pointing at the bird with a laser pointer to ensure that all three observers selected the same bird. The effectiveness of stunning was assessed in two different places of the slaughter line: (1) at the exit of the waterbath before bleeding; and (2) during bleeding at approximately 10 s after severing the carotids (Figure 1) in a representative sample of birds in each batch. The three observers evaluated each bird for 3 to 6 s (depending on slaughterhouse design and visibility) and took ABI scores. Observers assessed the ABIs independently and did not discuss or disclose their assessments during the evaluation.

#### 2.3.2. Sample Assessment

All batches of broiler chickens slaughtered in the presence of the observers in the plant were evaluated. In each batch, samples of 50–100 birds were assessed per cycle before and during bleeding. In order to get the largest possible sample, the cycle was repeated until the whole batch had been slaughtered.

Sometimes an observer was distracted for whatever reason (e.g., business operators passing in front of them) and failed to score the broiler chicken that was being indicated with the laser. In these cases, the observers made a note in their observations and the outcomes were not used for repeatability assessments.

A summary of the electrical parameters used for each batch and SH, along with the characteristics of the animals in the batch and the number of assessed birds, is shown in Table 2.

#### 2.3.3. Indicators for the Assessment

The ABIs for the assessment of the state of consciousness before and during bleeding were adapted from those proposed by the EFSA [6]. The selected ABIs before bleeding were tonic seizure, breathing, spontaneous blinking and vocalizations, while the selected ones during bleeding were wing flapping, breathing, spontaneous swallowing and head shaking. The description and outcome regarding consciousness and unconsciousness of these ABIs are summarized in Table 3.

The three trained observers agreed beforehand on the definition of the indicators, the methodology of assessment and the scoring to standardize the protocol when assessing the birds with these indicators.

Before the assessments of the birds, the three assessors were placed in a position where there was the best possible visibility of the shackled birds from a ventral position. However, due to divergence in the design and construction of the SH, sometimes the birds were assessed from a dorsal instead of a ventral position (SH-5 and SH-6 at the exit of the waterbath and in SH-3 during bleeding), thus impairing assessments of breathing by direct observation of abdominal muscles around the cloaca rhythmic movements. Data were recorded as binomial, i.e., as 0 if consciousness was not observed and 1 when consciousness was observed. The presence of at least one outcome of consciousness may indicate that the bird is conscious or regaining consciousness after WBS, and therefore, may indicate an ineffective stunning or a long stun-to-stick interval (i.e., time from stunning to the start of bleeding).

### 2.4. Statistical Analysis

Data pre-processing, statistical analyses and plots were performed using R software v.4.1.0. [15]. First, birds that were not assessed by all three observers were filtered out to ensure that all observations were directly comparable. For all statistical analyses, significance was declared at *p* < 0.05.

#### 2.4.1. Inter-Observer Repeatability of Animal-Based Indicators

The overall level of agreement between observers for each ABI was determined and expressed by the crude proportion of agreement (PoA) and the Fleiss’ kappa (κ) using the “irr” package of R software [16]. The PoA can be misleading, as it does not take into account the scores that the raters assign due to chance. κ overcomes this issue, as it provides an inter-observer agreement measure between two or more observers when the variable assessed is on a binomial or categorical scale. It expresses the degree to which the observed proportion of agreement among observers exceeds what would be expected if all observers made their ratings completely randomly. κ can range from −1 to +1, where 0 indicates the amount of agreement that can be expected from random chance, and 1 represents perfect agreement between the observers [17]. κ is a standardized value, and thus, is interpreted the same across multiple studies. According to Fleiss et al. [18] κ can be classified as “excellent” agreement beyond chance if values are greater than 0.75; “fair to good” agreement beyond chance with values between 0.40 and 0.75; and “poor” agreement beyond chance if values are below 0.40. However, when there is an insufficient scoring variation in a given indicator (i.e., a low prevalence of indicators of consciousness) but high agreement among observers, κ appears close to 0.

#### 2.4.2. Correlation among Animal-Based Indicators

The Chi-squared % defective test was used to determine if there were statistical differences among observers between the expected and the observed frequencies of every outcome of the evaluated indicators. If one observer differed statistically from the others in his/her evaluations of the ABIs, the mean of the proportion of the two closest evaluations or the in-between value when scoring was not consistent. In such cases, the association between the observed ABIs was determined using Spearman’s rank correlation, as these data did not follow a normal distribution. Correlation results are displayed as heat maps. Proportions among combinations of ABIs were illustrated as a Venn diagram considering all broilers assessed in the present study using the “eulerr” package [19].

## 3. Results

The ABIs were assessed on a total of 2685 broilers before bleeding and 3154 during bleeding from six different SH in France and Spain. However, not all of them were assessed by the three observers. Those not assessed by all three observers were filtered out. Thus, 2608 broilers remained in the dataset before bleeding and 3105 during bleeding. The number of the birds assessed per SH as well as the number and percentage of birds assessed by the three observers are shown in Table 4.

### 3.1. Inter-Observer Repeatability of the Animal-Based Indicators

#### 3.1.1. Before Bleeding

After WBS and before bleeding, four ABIs regarding the state consciousness were assessed: tonic seizure breathing, spontaneous blinking and vocalization. The prevalence of birds showing outcomes of consciousness by observer and SH is shown in Table 5. The overall level of agreement between the three observers for these ABIs according to the SH is shown in Table 6.

##### Tonic Seizure

Birds showing an absence of tonic seizure at the exit of the waterbath were observed in SH 1, 2, 3 and 6. However, there was divergence in the prevalence of this according to the SH assessed (Table 5). While SH-3 did not exceed 0.5% on average among observers, SH-6 had the highest prevalence in the sample, with an average of 36.6%. The PoA was above 91% in all SH except for SH-6, where it was lower (61%) due to divergences in the scoring (Table 6). Observer C scored 1.4 times more birds with absence of tonic seizure compared with the other observers (*p* < 0.0001; Table 5). Moreover, κ strongly varied among SHs: SH-1, SH-4 and SH-5 were close to 0, and as such, were interpreted as being in “poor agreement”, while SH-2, SH-3 and SH-6 were between 0.47 and 0.65 and were interpreted as “fair to good” (Table 6). A κ close to 0 reflects an insufficient scoring variation linked to a low prevalence of birds showing an absence of tonic seizure.

Considering the data from the total number of birds assessed in the present study (*n* = 2608), 7.5% of birds showed an absence of tonic seizure (Table 5), the PoA among observers was high (91.7%) and κ was statistically significant and interpreted as “fair to good” (κ = 0.64; *p* < 0.001; Table 6).

##### Breathing

Birds with the presence of breathing were observed in SH-1, SH-2 and SH-6 (Table 5). The highest prevalence of breathing was found in SH-1, with an average of 11.4%. The PoA was above 93% in all SHs (Table 6) and there was no divergence in ratings among observers (*p* > 0.05) in any SH nor batch (Table 5). However, there was divergence of κ linked to the different degree of prevalence of breathing among SHs (Table 6).

Taking all birds from the assessed SHs into consideration, the presence of breathing was observed in 0.9% of birds (Table 5), the PoA among observers was high (98.9%) and the κ was statistically significant and interpreted as “fair to good” (κ = 0.58; *p* < 0.001; Table 5).

##### Spontaneous Blinking

Birds showing spontaneous blinking were observed in SH-1 (0.9%) and SH-5 (0.2%), and there was no divergence in ratings among observers (*p* > 0.05) in any SH (Table 5). The PoA was above 98.0% in all SH but there was no divergence of κ, which was usually close to 0 (Table 6), showing that the prevalence of spontaneous blinking was low, considering the number of birds assessed.

It should be highlighted that spontaneous blinking was observed in 0.1% of the total birds assessed (Table 5) with a high PoA among observers (more than 99.8%) but with low κ; which this was statistically significant, it was interpreted as “poor” agreement (k = 0.14; *p* < 0.001; Table 5).

##### Vocalization

Vocalization was heard only in SH-5 with an average of 0.2% of the broilers (Table 5). Among all ABIs assessed before bleeding, vocalization was the one with the highest PoA (above 99.8%) and there was no divergence in ratings among observers (*p* > 0.05) in all SH assessed (Table 6).

Taking all birds assessed in this study into consideration, the detection of vocalization was extremely low (0.04%; Table 5); the PoA among observers was 100% but the κ was not computed due to insufficient scoring variation (Table 6).

#### 3.1.2. During Bleeding

Four ABIs were evaluated during bleeding: wing flapping, breathing, spontaneous swallowing and head shaking. The prevalence of birds showing indicators of consciousness according to the ABI per observer and SH assessed is shown in Table 7. In addition, the overall level of agreement between the three observers for these ABIs according to the SH is shown in Table 8.

##### Wing Flapping

Birds with presence of wing flapping were observed in SH-2 (0.6%), SH-5 (1.4%) and SH-6 (5.0%), and there was uniformity in the ratings among observers (*p* > 0.05) in all SHs except for the broilers assessed at SH-6, where one observer scored 2.3 times more wing flapping than the others (*p* < 0.001), as shown in Table 7. Therefore, the prevalence of wing flapping strongly differed between these SHs and thus, the κ and its interpretation ranged from “poor” to “fair to good” agreement, although the PoA among observers was above 94% in all SHs (Table 8).

Taking all birds from the assessed SHs into consideration, the detection of wing flapping differed statistically among evaluators (*p* < 0.01) and thus, the prevalence was considered to be 1.6%, as this was the in-between value (Table 7). However, the PoA among observers was high (98.2%) and the κ was statistically significant and interpreted as showing “fair to good” agreement (κ = 0.66; *p* < 0.001; Table 8).

##### Breathing

Birds with the presence of breathing during bleeding were observed in all SHs assessed except SH-4. The highest prevalence occurred in SH-6 (36.9%), followed by SH-2 (15.7%), SH-1 (4.1%), SH-3 (1.7%) and SH-5 (0.4%). Divergence on rating breathing among observers occurred at SH-3 and SH-6, as shown in Table 7.

Considering the data from all SH, breathing was observed in 13.6% of the assessed birds (Table 7). The average PoA was 88.2% and the κ was statistically significant and interpreted as showing “fair to good” agreement (κ = 0.64; *p* < 0.001; Table 8).

##### Spontaneous Swallowing

Birds showing spontaneous swallowing were observed in SH-1 (0.5%), SH-2 (1.9%) and SH-6 (0.7%). However, its prevalence was low compared with the presence of other outcomes of consciousness, and there was no divergence in the scoring of this parameteramong observers (*p* > 0.05), as shown in Table 7.

For this reason, there was no divergence of κ and the level of agreement was classified as “poor” under every condition tested (Table 8) [18].

Among all the assessed birds, the prevalence of spontaneous swallowing was 0.7% (Table 7) and the PoA was 96.4%, but the κ was low but statistically significant and interpreted as showing “poor” agreement (κ = 0.20; *p* < 0.001; Table 8).

##### Head Shaking

Birds showing head shaking were observed in SH-1 (0.5%), SH-2 (7.1%), SH-3 (0.9%), SH-5 (6.3%) and SH-6 (3.4%), and there was agreement at scoring head shaking among observers (*p* > 0.05), as shown in Table 7. Hence, variation in prevalence led to κ and levels of agreement which could be classified from “fair to good” to “excellent”, according to the SH assessed (Table 8).

Among all the assessed birds, head shaking was found in 3.8% (Table 7), the PoA among observers was 97.2%, and κ was statistically significant and interpreted as showing “fair to good” agreement (κ = 0.64; *p* < 0.001; Table 8).

### 3.2. Correlation among Animal-Based Indicators

#### 3.2.1. Before Bleeding

To elucidate the correlation among the outcomes of the ABIs assessed before bleeding, a contingency table was created. The proportions of birds showing outcomes of consciousness and combinations thereof observed in the same bird at this stage are shown as a Venn diagram in Figure 2a. An absence of tonic seizure was the most frequent indicator, followed by breathing. Spontaneous blinking, vocalization and combinations of the various outcomes of consciousness of the four ABIs were almost non-existent at this stage. A heat map was not generated for these data, since no correlation was found among any ABI.

#### 3.2.2. During Bleeding 

The proportion of birds showing outcomes of consciousness and their combinations observed at an individual level at this stage are shown as a Venn diagram in Figure 2b. This diagram shows that the presence of breathing was the most frequently observed outcome of consciousness, followed by head shaking and spontaneous swallowing, whereas the observation of wing flapping was rare. Additionally, when the prevalence of birds showing indicators of consciousness was high, as occurred at SH-2 and SH-6, some birds showed breathing accompanied primordially by head shaking, but rarely by spontaneous swallowing.

Unlike the before bleeding assessment, there was correlation among ABIs, as shown in the heat map in Figure 3. All correlations were positive but some were statistically significant, such as the presence of breathing and wing flapping (r = 0.71; *p* < 0.001), breathing and head shaking (r = 0.90; *p* < 0.001) and head shaking and spontaneous swallowing (r = 0.63; *p* = 0.02). However, there was no correlation between wing flapping and spontaneous swallowing (r = 0.22; *p* = 0.337).

## 4. Discussion

One of the aims of this study was to gain insights into the inter-observer repeatability of valid and feasible ABIs for the state of consciousness after WBS in broiler chickens. In addition, the prevalence of the outcomes of consciousness of the ABIs and correlations among them were computed in order to give insights into their reliability and relationship with the state of consciousness. This study compared the assessments of three observers of 5241 broilers from 19 batches in six different slaughterhouses and 11 different key electrical parameters applied in waterbaths from two main broiler producer countries in the EU. Even though the SHs were not randomly sampled, they represented a diverse range of equipment designs, key electrical parameter combinations and line speeds. In addition, it should be highlighted that not were the only observers well trained, but they also agreed on the definition of the indicators before assessing the birds. The number and position of the observers was intended to result in minimal interference to the operators. Although there was a restriction in terms of the available space for assessments, the observers stood next to each other, assessing the same animals at the same time.

### 4.1. Inter-Observer Repeatability of Animal-Based Indicators

Data were analyzed at individual broiler level and the combination of κ and PoA was used to assess the inter-observer repeatability of the outcomes of some ABIs regarding the state of consciousness. This repeatability among observers could be interpreted as poor to excellent, based on to the calculated κ value [18]. Our results showed that for most of the indicators, the κ interpretation varied according to the SH assessed. This was mainly because κ values are strongly influenced by prevalence, and this differed among SHs; the lower the prevalence, the higher the agreement among observers (as all observers agreed on the absence of ABIs) and the poorer the interpretation of κ. The higher the prevalence, the higher the likelihood of disagreement. The only exceptions to this were in the assessment of spontaneous swallowing and spontaneous blinking, where κ was interpreted as showing poor agreement among observers in all cases. In contrast, regarding vocalizations, the κ could not be computed due to a lack of outcomes of consciousness of this indicator. These results suggest that when prevalence is low, the calculation of κ does not give much information. A similar outcome occurs when paying attention to the PoA. A high PoA may suggest that there is a high level of agreement among observers. However, it may happen that the agreement is high because the outcome of consciousness of the indicator is very clearly detectable for all when present (e.g., head shaking), or because the outcome of consciousness is rarely (e.g., spontaneous blinking and swallowing) or hardly ever observed (e.g., vocalizations). It should be noted that the level of noise in the SHs did not impair the assessments of vocalizations. This was so because vocalizations are high in pitch and were clearly detectable before the broilers entered the waterbath and were still conscious. However, vocalizations after waterbath stunning were only detected in one out of 2608 broilers, and in this instance, all the three observers were in agreement.

Inter-observer repeatability in broilers is high for some ABIs regarding state of consciousness after WBS. The most repeatable indicator before bleeding is vocalization and spontaneous blinking, followed by tonic seizure and breathing. However, spontaneous blinking and vocalization were artificially highly repeatable because they were seldom observed. When considering these results, we recommend keeping tonic seizure and breathing at this stage, despite their being less repeatable among observers.

The most repeatable indicators during bleeding were found to be wing flapping, head shaking and spontaneous swallowing followed by breathing. Nevertheless, spontaneous swallowing and wing flapping were artificially highly repeatable because they were observed on few occasions. While the span of observation during bleeding was set from 10 s to 16 s post-neck cutting, some birds started to flap their wings at the end of this time. This generated doubts in terms of scoring and affected the consensus among the observers when wing flapping was present. This reflects the importance of setting an optimal observation duration where more accurate outcomes of consciousness may be observed within a slaughterhouse during bleeding. Additionally, but at lower scale, sometimes there were difficulties in differentiating wing flapping from movements of the wings caused by line shaking. Despite this, we recommend keeping breathing, wing flapping and head shaking as key ABIs during bleeding based on their high prevalence and repeatability.

Repeatability among the three observers could be influenced by impaired visibility of the animals because of the slaughterhouse design or because, when paying attention to a specific ABI, the evaluator is more prone to miss a positive outcome of another. However, it is likely that higher levels of inter-observer reliability could be achieved by standardizing descriptions and through training and wider testing. Hence, adequate training appears to be one of the key points to improve in animal welfare assessments of SHs. This can be achieved through theoretical lectures and video clips designed to train assessors, as well as practical field exercises in SHs until trainees harmonize their scoring with those done by experts [20].

### 4.2. Correlation among Animal-Based Indicators

Based on our observations, pre-stun shocks or runt (small) animals could have been the cause of non-stunned birds at the exit of the waterbath due to a lack of contact of the head with the electrified water. This explains the presence of broilers that remained conscious and showed combinations of indicators of consciousness before bleeding. On the other hand, some birds did not exhibit tonic seizure, and this indicator was not correlated to other outcomes of consciousness before bleeding. This may have been because the tonic seizure occurred while the bird was submerged in the waterbath, as might happen in long baths or with slow line speeds. On the other hand, it is known that when the electrical parameters are set to stun-to-kill the birds, the induction of cardiac arrest leads to reduced or an absence of tonic seizure at the exit of the waterbath [14], and does not mean that birds are conscious. In this sense, tonic seizure might not be as reliable as the other indicators of consciousness, since it depends on SH configuration and the current delivered.

Data on the order of the re-appearance of indicators during recovery in poultry are not described in literature. Despite the importance of these indicators, in the context of a slaughterhouse, their precise relationships with the brain state or with other indicators of consciousness are insufficiently known. Study of the relationships between different ABIs of state of consciousness may benefit from analyses by correlation [21]. In the present study, 14% of birds with at least one outcome of consciousness during bleeding showed simultaneously more than one outcome of consciousness. The most observed indicator of consciousness during bleeding was the presence of breathing, indicating a return of consciousness. It seems that when a broiler starts breathing, it is more prone to show movements such as head shaking and/or wing flapping later in the line.

Taking all this into consideration, assessing if the broilers are breathing is the most recommended ABI among those assessed in the present study after WBS and before bleeding. However, wing flapping, although not included, should be also considered before bleeding, as some birds were suspected of missing the waterbath (and therefore were not stunned at all) flapped their wings. During bleeding, breathing is the most observed indicator of consciousness; when observed, it was sometimes accompanied by wing flapping and/or head shaking.

## 5. Conclusions and Recommendations

Before bleeding, breathing and wing flapping are the most relevant and commonly observed indicators of consciousness, whereas spontaneous blinking and vocalizations may be considered secondary parameters. Any broilers that showed at least one of these ABIs should be re-stunned with backup stunning equipment before performing the neck cut, while the shackling of runts should be avoided, since they are more prone to skipping the waterbath.

During bleeding, breathing, wing flapping and head shaking are the most relevant and commonly observed ABIs of consciousness. Like before bleeding, any broiler that shows at least one outcome of consciousness at this stage should be re-stunned with backup stunning equipment as soon as it is detected. If some broilers of the same batch regain consciousness, the electrical parameters should be readjusted to ensure that all birds remain unconscious until death.

The repeatability in terms of evaluating the consciousness of broilers is likely to be improved by standardizing descriptions, robust training and wider testing in slaughterhouses.

This work will proposed a refined and validated list of indicators for use in assessments of the consciousness of broiler chickens in commercial slaughterhouses.

## Figures and Tables

**Figure 1 animals-12-01800-f001:**
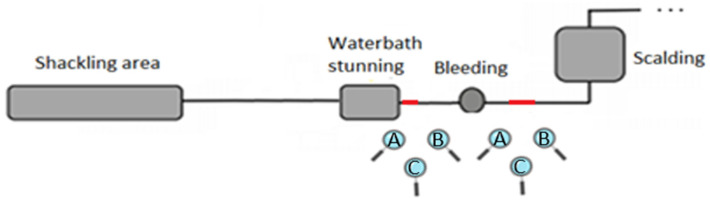
Position of the observers (A to C) during the assessment of ABIs in terms of the effectiveness of waterbath stunning in broilers. The position of the lens is the position of the observers (i.e., before and during bleeding) and the red segments are the observation areas.

**Figure 2 animals-12-01800-f002:**
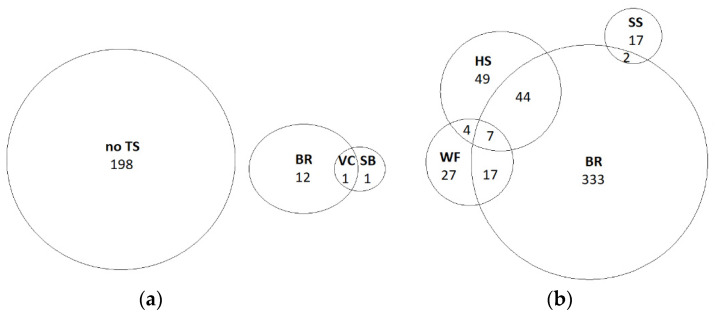
Venn diagram of animal-based indicators of consciousness assessed in broilers (**a**) before bleeding in waterbath stunned broilers and (**b**) during bleeding. Indicators of consciousness were: no TS: absence of tonic seizure; BR: presence of breathing; SB: presence of spontaneous blinking; VC: presence of vocalization; WF: presence of wing flapping; HS: presence of head shaking; SS: presence of spontaneous swallowing. Numbers specify the total number of broilers showing each indicator or combinations of indicators from a total of 2608 broilers assessed before bleeding and 3105 during bleeding.

**Figure 3 animals-12-01800-f003:**
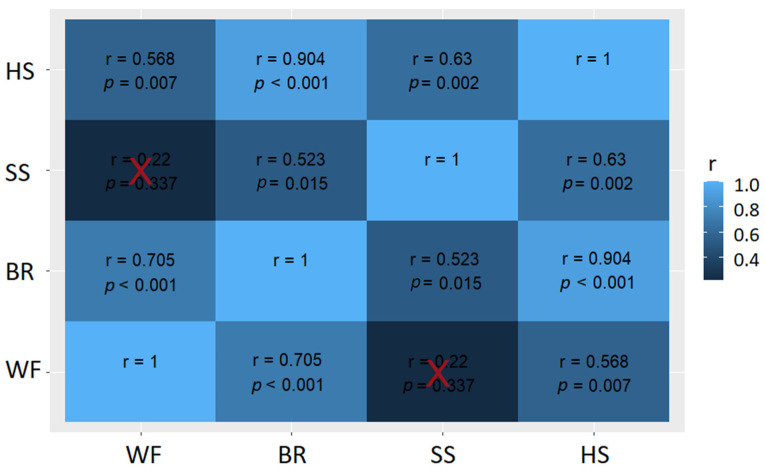
Heat map of correlations of the outcomes of the animal-based indicators regarding the state of consciousness during bleeding in waterbath stunned broilers (WF: presence of wing flapping; BR: presence of breathing; SS: presence of spontaneous swallowing; HS: presence of head shaking) observed when assessing 19 batches from six different slaughterhouses. The values in the table are Spearman’s rank correlation coefficients (r) and *p*-values. Red crosses indicate non-significant correlations (*p* > 0.05).

**Table 1 animals-12-01800-t001:** Main characteristics of the six slaughterhouses (1 to 6) included in the study.

	Slaughterhouse					
	1	2	3	4	5	6
Location	France	France	France	Spain	Spain	Spain
Waterbath length (m)	0.9	6.0	3.0	3.7	3.8	3.3
Birds in the waterbath (n)	3	39	11	16	18	12–14
Exposure time (s)	30	14	9	13	15	11
Line speed (birds/h)	200	9500	10,500	6000	6100	6000
Mean time from the exit of the waterbath until bleeding (s)	1	2	3	9	11	6
Bleeding method *	M	A	A	AM	AM	M

* Bleeding method: M (manually); A (automatically); AM (combination of first automatically and afterwards manually).

**Table 2 animals-12-01800-t002:** Number of batches of slaughtered broilers, type of strain, average body weight and age of broilers per batch for each slaughterhouse. The number of broilers assessed before and during bleeding and the average electrical parameters ± standard deviation of the waterbath are reported.

		Characteristics of the Birds				No. Birds Assessed		Electrical Parameters		
SH	Batch	Strain	No. Birds	BW, kg	Age, d	BB	DB	Current, mA/bird	Frequency, Hz	Voltage, V
1	1	SG	95	2.8	134	55	50	102 ± 15	60	80
	2	SG	57	2.7	107	46	226	105 ± 26	60	80
	3	SG	311	2.1	104	161	161	112 ± 29	60	80
2	1	FG	37,047	2.100	38	200	65	228 ± 33	793	211
	2	FG	23,647	2.260	37	200	200	273 ± 53	792	208
	3	FG	18,465	2.140	38	150	39	226 ± 64	607	209
	4	FG	11,693	1.835	37	200	239	177 ± 62	568	194
	5	SG	7280	1.860	37	200	200	234 ± 46	607	209
	6	SG	7280	1.860	37	50	50	352 ± 33	1500	279
3	1	FG	8000	1.860	34	200	128	157 ± 7	110	105
	2	FG	14,300	1.920	34	200	401	162 ± 8	110	95
4	1	FG	3240	2.843	44	104	0	309 ± 43	347 ± 1	68 ± 13
	2	FG	3888	2.837	44	100	124	276 ± 71	348 ± 1	60 ± 22
	3	FG	3.888	2.924	44	50	150	255 ± 59	349 ± 1	65 ± 17
5	1	FG	1458	2.092	35	41	0	224 ± 0	352 ± 1	195 ± 12
	2	FG	5832	2.047	35	200	234	223 ± 3	352 ± 1	198 ± 12
	3	FG	2916	1.763	35	100	131	223 ± 0	352 ± 1	206 ± 13
	4	FG	2916	1.763	35	100	150	224 ± 0	352 ± 1	215 ± 14
6	1	FG	2700	3.560	56	100	0	105 to 141	80	33
	2	FG	720	4.020	60	0	38	105 to 111	80	32
	3	FG	4350	3.570	55	0	200	105 to 143	80	32
	4	FG	6300	2.830	43	100	195	108 to 153	80	32
	5	FG	5040	3080	50	100	100	106 to 160	80	33

SH: slaughterhouse; No. Birds: number of birds in the batch; BW: body weight; SG: slow growing; FG: fast growing; BB: before bleeding; DB: during bleeding.

**Table 3 animals-12-01800-t003:** Animal-based indicator (ABI) assessments and descriptions of the outcomes of unconsciousness and consciousness in broilers stunned by waterbath in two different stages: before and during bleeding. Adapted from EFSA [15].

Stage	ABI	Outcome of Unconsciousness	Outcome of Consciousness
Before bleeding	Tonic seizure	Bird shows general loss of muscle tone and a completely relaxed and flaccid body, with no neck tension.	Bird shows arched and stiff neck (i.e., necks appear parallel to the ground) and wings held tightly close to the body.
	Breathing	Absence of movements of the beak or abdominal muscles around the cloaca associated with the cessation of breathing.	Occurrence of a minimum of two movements of either the beak or abdominal muscles around the cloaca, associated with breathing.
	Spontaneous blinking	Bird does not open/close eyelid on its own (fast or slow) without stimulation.	Bird opens/closes eyelid on its own (fast or slow) without stimulation.
	Vocalizations	Absence of single or repeated short and loud shrieking (screaming) at high frequencies.	Single or repeated shrieking (screaming).
During bleeding	Wing flapping	Absence of flapping with both wings.	Flapping with both wings; this should not be confused with rapid trembling of the entire body of the bird.
	Breathing	Absence of movements of the beak or abdominal muscles around the cloaca associated with the cessation of breathing.	Occurrence of a minimum of two movements of either the beak or abdominal muscles around the cloaca, associated with breathing.
	Spontaneous swallowing	Absence of deglutition reflex.	Deglutition reflex triggered by water from the stunner or blood from the neck-cutting wound entering the mouth during bleeding.
	Head shaking	Bird does not shake its head from side to side.	Bird shakes its head from side to side to get rid of blood or water entering the nostrils.

**Table 4 animals-12-01800-t004:** Number of assessed animals and number and percentage of birds that were able to be assessed by the three observers according to the slaughterhouse (before and during bleeding).

	Slaughterhouse						
Birds Assessed	1	2	3	4	5	6	All
Before bleeding							
Total number of birds assessed	189	984	400	254	441	417	2685
Number of birds assessed by the three observers	114	984	400	254	441	415	2608
Birds assessed by the three observers, %	60.3	100.0	100.0	100.0	100.0	99.5	97.1
During bleeding							
Total number of birds assessed	209	793	529	374	516	733	3154
Number of birds assessed by the three observers	195	778	527	374	515	716	3105
Birds assessed by the three observers, %	93.3	98.1	99.6	100.0	99.8	97.7	98.4

**Table 5 animals-12-01800-t005:** Percentage of the outcomes of the animal-based indicators for the state of consciousness in broilers after waterbath stunning but before bleeding according to the observer (A to C) and slaughterhouse (SH) assessed.

		Absence of TS, %					Presence of BR, %					Presence of SB, %					Presence of VC, %				
SH	Birds, n	A	B	C	Mean	*p*-Value	A	B	C	Mean	*p*-Value	A	B	C	Mean	*p*-Value	A	B	C	Mean	*p*-Value
1	114	2.6	6.1	0.9	3.5	0.07	12.3	11.4	11.4	11.4	0.97	0.9	0.9	0.0	0.9	0.61	0.0	0.0	0.0	0.0	-
2	984	3.0	2.6	2.1	2.6	0.44	0.7	0.3	0.2	0.4	0.17	0.2	0.0	0.0	0.0	0.37	0.0	0.0	0.0	0.0	-
3	400	0.8	0.3	0.3	0.5	0.45	0.0	0.3	0.0	0.0	0.39	0.0	0.0	0.0	0.0	-	0.3	0.0	0.0	0.0	0.37
4	254	0.0 ^b^	0.8 ^a^	0.0 ^b^	0.0	0.05	0.0	0.0	0.0	0.0	-	0.0	0.0	0.0	0.0	-	0.0	0.0	0.0	0.0	-
5	441	3.4 ^a^	0.0 ^b^	0.0 ^b^	0.0	<0.01	0.2	0.2	0.2	0.2	1.00	0.2	0.0	0.2	0.2	0.61	0.2	0.2	0.2	0.2	1.00
6	415	36.4 ^b^	36.6 ^b^	59.8 ^a^	36.6	<0.01	0.7	1.4	1.0	1.0	0.58	0.0	0.0	0.0	0.0	-	0.0	0.0	0.0	0.0	-
All	2608	7.7 ^b^	7.2 ^b^	10.4 ^a^	7.5	<0.01	1.0	0.9	0.8	0.9	0.74	0.2	0.0	0.1	0.1	0.37	0.1	0.0	0.0	<0.1	0.78

TS: absence of tonic seizure; BR: presence of breathing; SB: presence of spontaneous blinking, VC: presence of vocalization; n: number of birds. a–b = Values with different superscripts within the same raw differ among observers by chance (*p* < 0.05).

**Table 6 animals-12-01800-t006:** Inter-observer proportion of agreement (PoA), Fleiss’ kappa coefficient (κ) and interpretation and standard error (SE) of the animal-based indicators for the state of consciousness before bleeding, according to the slaughterhouse assessed.

				Slaughterhouse			
Item	1	2	3	4	5	6	All
Tonic seizure							
PoA, %	91.2	97.4	99.5	99.2	96.6	61.0	91.7
Fleiss’ κ (SE)	0.06 (0.05)	0.65 (0.02)	0.60 (0.03)	0.00 (0.04)	−0.01 (0.03)	0.47 (0.03)	0.64 (0.01)
κ interpretation	Poor	Fair to good	Fair to good	Poor	Poor	Fair to good	Fair to good
p-value	0.131	<0.0001	<0.0001	0.529	0.700	<0.0001	<0.0001
Breathing							
PoA, %	93.0	99.1	99.8	100	100	97.4	98.9
Fleiss’ κ (SE)	0.77 (0.05)	0.25 (0.02)	0.00 (0.03)	*	*	0.14 (0.03)	0.58 (0.01)
κ interpretation	Excellent	Poor	Poor	*	*	Poor	Fair to good
*p*-value	<0.0001	<0.0001	0.512	*	*	<0.0001	<0.0001
Spontaneous blinking							
PoA, %	98.3	99.7	100	100	99.8	100	99.8
Fleiss’ κ (SE)	0.00 (0.05)	0.00 (0.02)	*	*	0.05 (0.03)	*	0.14 (0.01)
κ interpretation	Poor	Poor	*	*	Poor	*	Poor
*p*-value	0.540	0.522	*	*	<0.0001	*	<0.0001
Vocalisation							
PoA, %	100	100	99.8	100	100	100	100
Fleiss’ κ (SE)	*	*	*	*	*	*	*
κ interpretation	*	*	*	*	*	*	*
*p*-value	*	*	*	*	*	*	*

* Insufficient scoring variation to calculate κ coefficients (all indicator scores were 0). κ interpretation: ≥0.75 ‘excellent’, 0.40–0.74 ‘fair to good’, and < 0.40 ‘poor’ agreement [18].

**Table 7 animals-12-01800-t007:** Percentage of the outcomes of the animal-based indicators for the state of consciousness in waterbath stunned broilers during bleeding according to observer (A to C) and slaughterhouse (SH).

		Presence of WF, %					Presence of BR, %					Presence of SS, %					Presence of HS, %				
SH	Birds, n	A	B	C	Mean	*p*-Value	A	B	C	Mean	*p*-Value	A	B	C	Mean	*p*-Value	A	B	C	Mean	*p*-Value
1	195	0.0	0.5	0.0	0.0	0.13	4.6	5.1	3.1	4.1	0.39	0.5	0.5	1.0	0.5	0.86	0.5	1.0	0.5	0.5	0.86
2	778	1.2	0.5	0.3	0.6	0.17	15.9	16.1	15.0	15.7	0.89	2.8	1.0	1.8	1.9	0.07	6.9	8.5	5.9	7.1	0.21
3	527	0.0	0.2	0.0	0.0	0.37	1.9 ^a^	0.2 ^b^	1.3 ^a^	1.7	0.01	0.0	0.2	0.0	0.0	0.37	0.9	0.9	0.9	0.9	1.00
4	374	0.0	0.0	0.0	0.0	-	0.0	0.0	0.0	0.0	-	0.0	0.0	0.0	0.0	-	0.0	0.0	0.0	0.0	-
5	515	1.4	1.2	1.4	1.4	0.96	0.8	0.4	0.2	0.4	0.19	0.0	0.0	0.0	0.0	-	0.0	0.0	0.0	0.0	-
6	716	8.1 ^a^	5.3 ^b^	4.6 ^b^	5.0	<0.01	36.9 ^b^	41.1 ^a^	28.8 ^c^	36.9	<0.01	0.4	0.6	1.0	0.7	0.19	6.7	6.6	5.6	6.3	0.44
All	3105	2.4 ^a^	1.6 ^ab^	1.4 ^b^	1.6	<0.01	13.2 ^a^	13.9 ^a^	10.9 ^b^	13.6	<0.01	0.8	0.5	0.7	0.7	0.15	3.5	3.9	3.9	3.4	0.63

WF: wing flapping; BR: breathing; SB: spontaneous swallowing; HS: head shaking; n: number of birds. a–c = Values with different superscripts within the same raw differed among observers by chance (*p* < 0.05).

**Table 8 animals-12-01800-t008:** Inter-observer proportion of agreement (PoA), Fleiss’ kappa coefficient (κ) and its interpretation and standard error (SE) of the animal-based indicators for the state of consciousness during bleeding according to the slaughterhouse assessed.

Item	Slaughterhouse						
	1	2	3	4	5	6	All
Wing flapping							
PoA, %	99.5	98.6	99.8	100	99.6	94.3	98.2
Fleiss’ κ (SE)	0.00 (0.04)	0.26 (0.02)	0.00 (0.03)	*	0.00 (0.03)	0.66 (0.02)	0.66 (0.01)
κ interpretation	Poor	Poor	Poor	*	Poor	Fair to good	Fair to good
*p*-value	0.517	<0.0001	0.510	*	<0.0001	<0.0001	<0.0001
Breathing							
PoA, %	93.0	85.2	97.7	100	100	97.4	88.2
Fleiss’ κ (SE)	0.58 (0.04)	0.63 (0.02)	0.33 (0.03)	*	0.57 (0.03)	0.54 (0.02)	0.64 (0.01)
κ interpretation	Fair to good	Fair to good	Poor	*	Fair to good	Fair to good	Fair to good
*p*-value	<0.0001	<0.0001	<0.0001	*	<0.0001	<0.0001	<0.0001
Spontaneous swallowing							
PoA, %	98.5	99.5	99.8	100	100	98.5	98.4
Fleiss’ κ (SE)	0.24 (0.04)	0.19 (0.02)	0.00 (0.03)	*	*	0.21 (0.02)	0.20 (0.01)
κ interpretation	Poor	Poor	Poor	*	*	Poor	Poor
*p*-value	<0.0001	<0.0001	0.510	*	*	<0.0001	<0.0001
Head shaking							
PoA, %	99.5	92.9	99.4	100	100	92.6	96.4
Fleiss’ κ (SE)	0.75 (0.04)	0.64 (0.02)	0.80 (0.03)	*	*	0.58 (0.02)	0.64 (0.01)
κ interpretation	Excellent	Fair to good	Excellent	*	*	Fair to good	Fair to good
*p*-value	<0.0001	<0.0001	<0.0001	*	*	<0.0001	<0.0001

* Insufficient scoring variation to calculate κ coefficients (all indicator scores were 0). κ interpretation: ≥0.75 ‘excellent’, 0.40–0.74 ‘fair to good’, and <0.40 ‘poor’ agreement [18].

## Data Availability

Data is contained within the article.

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
