# Peer review of "Inter-Observer Repeatability of Indicators of Consciousness after Waterbath Stunning in Broiler Chickens"

_animals, 2022, doi:10.3390/ani12141800_

Round 1

Reviewer 1 Report

I think this is an interesting and important paper. My comments are all fairly minor

Line 22 English. Replace  “serve at proposing” with “propose”

Line 30. English.  Remove “For this,”

Line 32. English. Replace  ”observers in” with “observers of”

Line 32.  English. Replace “of six” with “in six”

Line 34. English. Replace “allowed to” with “allowed us to”

Line 34. English. Replace and neglect” with “and to neglect”

I will not continue with minor language corrections where they do not hinder understanding.

Lines 120   for clarity should you

·       use the word “automatically” rather than “mechanically”

·       say “first automatically and afterwards…. “?

·       change MA to AM and

Line 148 what is the 3-6 seconds? Duration of observation?,  interval between observation? Time since exiting the water bath?

Lines 156 – 159  I am not really sure what this means. Are you saying no results were rejected?... or no batches that you intended to assess were not assessed (because for example they were to easy? difficult? Etc.   

Lines 194-196   where there was uncertainty I guess that the observers may have hesitated and then fail to enter a value… ie are you biasing the results towards the clear indicators?

Section 2.4.2 Some observers will be more sensitive to particular indicators than others. The text suggests that you responded to this. Was this the best approach? You refer to ven diagrams and a heat map but do not show them.

Table 5. the line wrapping this and other tables makes them difficult to read. Could you format it differently?

Author Response

Line 22: English. Replace “serve at proposing” with “propose”: Done

Line 30: English.  Remove “For this,”: Done

Line 32: English. Replace “observers in” with “observers of”: Done

Line 32:  English. Replace “of six” with “in six”: Done

Line 34: English. Replace “allowed to” with “allowed us to”: Done

Line 34: English. Replace and neglect” with “and to neglect: Done

I will not continue with minor language corrections where they do not hinder understanding. We appreciate your effort at revising and correcting the manuscript. We have contacted a native speaker to revise the whole document.

Line 120:   for clarity should you

  • use the word “automatically” rather than “mechanically”: Done. Also changed every time that “mechanically” appeared from L133 to L136
  • say “first automatically and afterwards…. “?: Done
  • change MA to AM and: Done

Line 148: what is the 3-6 seconds? Duration of observation?, interval between observation? Time since exiting the water bath? Duration of observation. Changed with “The three observers evaluated each bird during 3 to 6 s depending on slaughterhouse design and visibility and scored the ABIs.”

Lines 156 – 159:  I am not really sure what this means. Are you saying no results were rejected? or no batches that you intended to assess were not assessed (because for example they were to easy? difficult? Etc. Not at all, we assessed as much broilers as we could. We were moving every 50 to 100 animals from during bleeding stage to before bleeding stage and viceversa until the whole batch was slaughtered. One of our goals was to evaluate the largest  sample size possible. Find additional explanation in L159-160.

Lines 194-196:  where there was uncertainty I guess that the observers may have hesitated and then fail to enter a value… ie are you biasing the results towards the clear indicators?. It never was a matter of hesitation. Sometimes an observer was distracted for whichever reason (e.g. business operators passing in front of us) and missed to score the chicken pointed with the laser. Clarified in L162-165.

Section 2.4.2: Some observers will be more sensitive to particular indicators than others. The text suggests that you responded to this. Was this the best approach? You refer to ven diagrams and a heat map but do not show them. The three observers have considerable training at detecting indicators of consciousness (stated in L141). However, sometimes an indicator can be missed due to several reasons (see L454-466). Despite the limitations, we thought that what the majority of the observers scored is supposed to be the closest approach to the truth. Venn diagrams and Heat map is shown in Figures 2 and 3, respectively.

Table 5. the line wrapping this and other tables makes them difficult to read. Could you format it differently? We moved the abbreviations in titles to the bottom of the tables. Hope it can be easier to read in this way.

Reviewer 2 Report

Review of the manuscript Animals-1765876, entitled
‘Inter-observer Repeatability of Indicators of Consciousness after Waterbath Stunning in Broiler Chickens’

for Animals

Manuscript is well written and reports new insight on the broiler chickens welfare assessment during the slaughter. The work is much valuable because it serve at proposing a refined and validated list of indicators so that they can be used to assess the consciousness of broiler chickens in commercial slaughterhouses.  

It was a great pleasure to review a thoughtful and well-structured manuscript. The problem is concisely stated, the methods are described comprehensively, and the conclusions are justified by the results. The paper is presented in clear form for evaluation to audience, coherent and compelling.

Specific comments

1. What is the main question addressed by the research?

The main question addressed by the research is if the examinated indicators (tonic seizure, breathing, spontaneous blinking and vocalisation, wing flapping, breathing, spontaneous swallowing and head shaking) are repeatable enough to be relevant in monitoring the state of consciousness in broiler chicken after waterbath stunning.

2. Do you consider the topic original or relevant in the field, and if so, why?

I consider the topic relevant in the field of animal welfare, because the aim of this research is to improve a list of indicators that can be used in commercial slaughterhouses ensuring consistency of assessments. It could avoid unnecessary pain, stress and suffering of broiler chickens connected with the weaknesses in monitoring of their state of consciousness after waterbath stunning.

3. What does it add to the subject area compared with other published material?

In comparison with other publications the evaluation of the inter-observer repeatability of the four most valid and feasible animal-based indicators  in different commercial slaughterhouses and batches of broilers. In addition, the prevalence of the outcomes of the animal-based indicators are calculated to gain insight on which of them are more prone to occur in case of ineffective stunning and the correlation among them into their reliability and relationship with the level of consciousness. With this, offer a reduced list of the most relevant ABIs so that they can be used to assess the consciousness of broiler chickens in commercial slaughterhouses ensuring consistency between observers.

4. What specific improvements could the authors consider regarding the methodology?

In my opinion the methods are described comprehensively and the methodology is well considered.

I suggest to improve Tables 5-8 so that particular values (e.g. P-value) are on one line.

5. Are the conclusions consistent with the evidence and arguments presented and do they address the main question posed?

Yes, they are. I recommend to shorten the section Conclusions and recommendations, because some sentences constitute a discussion. The authors should extract the most important conclusions and recommendations.

6. Are the references appropriate?

Yes, the references are works published in scientific journals from the last 20 years. According to the instructions for authors, the volume numbers should be in italics.

Author Response

I suggest to improve Tables 5-8 so that particular values (e.g. P-value) are on one line: Done

“I recommend to shorten the section Conclusions and recommendations, because some sentences constitute a discussion. The authors should extract the most important conclusions and recommendations”: Done

“According to the instructions for authors, the volume numbers should be in italics”: Done

Reviewer 3 Report

The manuscript deals with an important topic – the assessment of indicators of consciousness after stunning in broilers. This is a novel and important approach and the paper is well written and structured. However, I have some methological questions before a publication would be possible from my point of view.

You describe that the observers always stood at the same position. This means, that the position is a confounding variable – please consider this aspect in the results and especially in the discussion section. The exact genotypes of the broilers would be interesting, especially considering the slaughterhouse 1. Here also the slaughterhouse is confounded to genotype. Please consider this in the discussion. You do not mention genotype in the discussion (or the weight), however both should be included.

Did you measure the volume or loudness of the machines at slaughter? Were they different? This could have been one aspect why vocalization was not heard or only at an extremely low rate. Furthermore, vocalization is difficult to quantify. Please consider this in your methods, results and discussions section.

Detailed comments:

Line 46: you write that the head is submerged into a waterbath. Please be precise, according to the EU Legislation, the head has to be submerged until the clavicular bone.

Line 55: please state a reference for the thesis, that the electrical stunning is not always effective.

Line 283: please change to “Birds showing spontaneous blinking WERE observed”

Line 451: please change “head shacking” to “head shaking”

Table 2, Table 5 and Table 6:

Please use consistent decimal places and please format the tables in a way that numbers belonging to one unit (number) are is one row and not separated by line breaks.

Table 2: please state the standard deviation also for the SH 6 considering the frequency and use consistent decimal places.

Table 3: please use consistent wording for the same definitions (eg breathing). How often did the birds have to move their wing until you counted it as wing flapping?

Figure 2: please state the meanings of the colors in a foot note. Please change the A and B to a and b I the caption of the figure (according to the description of the figure).

Author Response

The exact genotypes of the broilers would be interesting, especially considering the slaughterhouse 1. Here also the slaughterhouse is confounded to genotype. Please consider this in the discussion. You do not mention genotype in the discussion (or the weight), however both should be included. We do have the information regarding the genotypes slaughtered in slaughterhouse 1. However, the genotype was different in each batch within the slaughterhouse 1. Since we would like to keep the anonymity of the slaughterhouses we are afraid that if we mention which genotypes were slaughtered, the country and the region may easily revealed. In addition, we agree that genotype/body weight affects the efficiency of stunning but it should not interfere on the repeatability of the indicators.

Did you measure the volume or loudness of the machines at slaughter? Were they different? We did not measure the pressure of sound in any slaughterhouse. We include this information in L138-139. This could have been one aspect why vocalization was not heard or only at an extremely low rate. Furthermore, vocalization is difficult to quantify. Please consider this in your methods, results and discussions section. Included in discussion section (L434-439).

Line 46: you write that the head is submerged into a waterbath. Please be precise, according to the EU Legislation, the head has to be submerged until the clavicular bone. Done: found in L46.

Line 55: please state a reference for the thesis, that the electrical stunning is not always effective. Reference added in L55.

Line 283: please change to “Birds showing spontaneous blinking WERE observed”. Done

Line 451: please change “head shacking” to “head shaking”. Done

Table 2, Table 5 and Table 6: Please use consistent decimal places and please format the tables in a way that numbers belonging to one unit (number) are is one row and not separated by line breaks. Done

Table 2: please state the standard deviation also for the SH 6 considering the frequency and use consistent decimal places. It was not possible to calculate the standard deviation of the frequency used in this slaughterhouse since the data retrieved from official veterinarian was not continuous in time but an interval per batch as shown in the following capture…

Table 3: please use consistent wording for the same definitions (eg breathing). Done. How often did the birds have to move their wing until you counted it as wing flapping?: We did not count how many times the birds flap their wings. We considered wing flapping as it was defined by EFSA (2013): “Flapping with both wings and should not be confused with rapid trembling of the entire body of the bird”

Figure 2: please state the meanings of the colors in a foot note. Unfortunately, there is no meaning… the colours are automatically selected by the R software when performing a Venn Diagram. In order to avoid confusion, Venn Diagrams have been replaced by another ones filled in blank. Please change the A and B to a and b I the caption of the figure (according to the description of the figure). Done.
